# Behavioral Welfare Research for the Management of Sanctuary Chimpanzees (*Pan troglodytes*)

**DOI:** 10.3390/ani13162595

**Published:** 2023-08-11

**Authors:** Amy Fultz, Rebekah Lewis, Liam Kelly, Jordan Garbarino

**Affiliations:** Chimp Haven, 13600 Chimpanzee Place, Keithville, LA 71047, USA; rlewis@chimphaven.org (R.L.); kellyl@jacksonvillezoo.org (L.K.); jgarbarino@chimphaven.org (J.G.)

**Keywords:** animal welfare, behavior, research, observations, behavioral management

## Abstract

**Simple Summary:**

This manuscript details seven different types of behavioral metrics utilized at Chimp Haven to monitor the welfare of the chimpanzees. Each metric focuses on a different aspect of chimpanzee behavior and well-being. Chimp Haven monitors the chimpanzees’ social behavior via wounding reports, Nearest Night Neighbor proximities, behavioral time budgets, and shifting logs. Individual chimpanzee welfare is monitored via hair loss and wellness surveys as well as targeted individual assessments for abnormal behavior and individual logs on a chimpanzee’s progress in our positive reinforcement training program. The metrics provided here may provide a framework for various types of objective and simple data collection for other animal organizations.

**Abstract:**

Chimp Haven is a sanctuary for chimpanzees retired from biomedical research, rescued from the pet trade, or re-homed after other organizations could no longer care for them. To provide optimal care for over 300 chimpanzees, Chimp Haven’s animal care team includes experts in behavioral science, veterinary treatment, and husbandry practices. To aid these teams in making routine welfare management decisions, a system of behavioral metrics provides objective data to guide decisions and track outcomes. Chimp Haven has built and piloted seven behavioral metric protocols over the past 5 years to provide staff with an objective and comprehensive picture of the chimpanzees’ behavioral welfare. The data from behavioral observations, staff surveys, and routine staff documentation are analyzed and processed through Google Forms, ZooMonitor, Microsoft Power Bi, Microsoft Excel, and R. Each metric assists staff in making data-based decisions regarding the management of captive chimpanzees related to abnormal behavior, hair loss, wounding, social relationships, positive reinforcement training and overall wellness. In this article, we explore examples of each metric and how they have been utilized to monitor and make decisions for both social groups of chimpanzees as well as individuals. These metrics can be collected and shared easily in an understandable format, which may provide an important framework for others to follow to enable the tracking of welfare for other sanctuaries, non-human primates, as well as other species.

## 1. Introduction

For over 30 years, there has been an increasing awareness of “animal welfare” in zoos, sanctuaries, agriculture, and research facilities [1]. Today, there is still a lack of consensus as to how to define what we mean by animal welfare [2]. In its broader understanding, animal welfare refers to the state of an individual in relation to their environment [3]. Animal welfare is challenging to measure as there is no single accepted welfare measure, and multiple indicators can and should be used to determine if an animal is in a good or bad welfare state. However, for primates, good welfare is generally considered to consist of good physiological health, nutrition, appropriate social structures, and the presence of species typical behaviors as well as the absence of abnormal or atypical behaviors [4,5,6]. With the advent of this awareness of animal welfare, there have been numerous studies of what we may consider “good” or “bad” welfare for a given individual primate [7,8,9]. In addition to these changes, there has also been an accompanying increase in employment opportunities in the animal welfare field and in fact, the field is projected to continue to grow [10]. One question that often arises when we discuss welfare is: how should welfare be measured? Should we only look at negative impacts on welfare or should we look at it in a more proactive and positive manner and determine what good welfare looks like as well [11,12]?

The Association of Zoos and Aquariums (AZA), the accrediting organization in the United States, now requires that every animal receive “a thorough yearly welfare assessment” [13]. Since 1986 [14], regulatory bodies of research facilities and other facilities inspected by the United States Department of Agriculture (USDA) have required a Behavioral Management Plan (for primates and dogs), which gives specific details about what enrichment will be provided, how positive reinforcement training will be conducted, as well as how animals that must be singly housed or cannot live in a social group will be treated, often by providing additional enrichment or time spent with staff [15,16]. However, there is no current requirement for behavioral welfare assessments or formal observational studies. 

Research, even non-invasive observational studies, in sanctuary settings, has been relatively uncommon in the past, even with species as long-lived as great apes who also require special consideration according to the Animal Welfare Act in the United States [15]. Although it has been historically uncommon in sanctuaries, this type of welfare focused research began in other managed settings such as research facilities and zoos. Sanctuaries often have restrictions on the types of studies that are allowed [17], and although the focus of the research is often applied behavioral management, the types of studies being conducted are becoming broader [18]. Only in the past 10 years or so has research become more common in sanctuaries that house great apes [19]. Some research has been conducted in chimpanzee (*Pan troglodytes*) sanctuaries in Spain and Japan, and a few studies have also been conducted in African sanctuaries. Japanese scientists have looked at cortisol levels and the effects of relocation [20] as well as social play as an indicator of positive welfare [21]. The Fundacio Mona Centre de Recuperacio de Primats sanctuary in Spain has focused on the rehabilitation and resocialization of their chimpanzees [22] as well as their development over time [23]. Chimfunshi, a sanctuary in Africa, has studied differences between chimpanzees removed from their mothers at early ages with those who were born and raised in the sanctuary [24]. A few North American sanctuaries have also conducted behavioral research on the chimpanzees in their care. Much of the research in sanctuaries in North America focuses on the social behavior of captive chimpanzees including integration of the chimpanzees [25], dominance hierarchies [26], and caregiver interactions with the chimpanzees [27,28]. Studies on enrichment [29], abnormal behavior [30], nesting [31,32,33], and their impacts on the welfare of sanctuary chimpanzees also exist throughout the sanctuary sector. This is by no means an exhaustive list of the types of research now conducted in sanctuary settings but is merely a representative sample of some of the welfare related work being conducted around the world (see Ross and Leinwand [19]) for a more in-depth look at sanctuary research. As this field of research continues to expand and develop, it is important to review various aspects of how sanctuaries utilize and apply data to validate ways to monitor the behavioral welfare of captive chimpanzees. 

Behavioral data collection can be time consuming, and it may require advanced training of staff members, specialized equipment, or even additional staff [34]. However, straightforward staff surveys have been shown to be effective and perform well when compared to more formal methods of data collection [11,35,36,37,38,39]. In a review of 10 years of personality research in non-human primates, 67% of 69 studies were based on caregivers familiar with the animals rating their personality traits [9]. Observer ratings on mobility and body condition scoring (BCS) in chimpanzees have also been verified as being accurate compared to observational studies (mobility) [40] and hands-on veterinary scoring such as body conditioning scoring (BCS) [41]. 

Chimp Haven has always strived to provide science-based care to the chimpanzees [42,43]. Although it is important to consider subjective information provided by those who care for the chimpanzees on a daily basis, evidence-based data are also important and should inform decisions that impact the care of the chimpanzees. At Chimp Haven, we have seven types of data-gathering metrics in place to inform the staff about varying aspects of the chimpanzees’ behavior. These metrics may also give us an avenue to find effective interventions or strategies to improve welfare. For each type of behavioral metric used, we provide examples from Chimp Haven’s population. Figure 1 shows the relationships between the different types of behavioral data collection, including how one protocol may lead to another being implemented. Data collection varies for each type of assessment, but each sampling method follows Altmann [44]. The first four assessments consist of regularly scheduled ongoing monitoring of the chimpanzees. The last three assessments are typically triggered by specific events or concerns that may impact the chimpanzees’ welfare. 

For the Wound-Tracking System, Social Monitoring System, Wellness Surveys, and Positive Reinforcement Training Logs, we utilize Microsoft Power Bi, which is a program that allows for the illustration and graphing of data which allows users to manipulate variables such as individual, group, timeframe, as well as other variables dependent on the protocol. These data are then routinely shared with animal care management and staff.

These seven methods of monitoring the chimpanzees allow us to look at the behavior of the colony at a specific point in time or during a particular date range. The metrics also allow for comparisons of wounding rates and hair loss over time, in different groups, or even as the chimpanzees are given access to novel spaces or experience different events in their lives. This allows us to monitor the chimpanzees at the individual, group and colony level. This paper provides examples of how various behavioral metrics can be easily measured in captive chimpanzees as well as in a variety of species. 

## 2. Facilities

Chimp Haven lies on 200 forested acres outside of Shreveport, Louisiana and consists of a variety of enclosures based on the individual needs of the chimpanzees. Enclosures include enclosed outdoor spaces with pine straw flooring; enclosed outdoor areas with a natural grass substrate, wire mesh walls, and solid metal brachiating bar ceilings that peak at a height of 7 m; 25-acre open air corrals with cement, wood and mesh walls; and large, multi-acre forested habitats bound by concrete, wood and mesh walls and water moats. Areas that are connected by multiple overhead mesh chutes enable the chimpanzees to be rotated throughout the various areas of the sanctuary. All chimpanzees have access to adjacent indoor bedrooms, excluding times when the areas are being cleaned or maintained. The chimpanzees also always have access to outdoor areas, unless there is need for maintenance or during times of inclement weather. All enclosures have elevated climbing structures and wire shelves, and the chimpanzees receive nesting materials of hay, pine straw, blankets or woodwool daily. The chimpanzees are fed a fresh produce diet and commercially available primate diet twice daily. Water is available ad libitum. Enrichment is provided daily, 365 days a year, and the chimpanzees receive additional grain forage at least three to four days a week.

## 3. Wellness Surveys

Wellness surveys are overarching surveys that allow us to gauge an individual chimpanzee’s behavioral wellness. Our goal is to observe each chimpanzee every three months for a total of four times each year. The quarterly time period allows us to observe individual chimpanzees during different seasons. This may assist us in determining if there are possible connections between the season and an individual chimpanzee’s behavior in the future. 

This survey utilizes Google Forms, which was distributed to select participants from Chimp Haven’s animal care team via email. The survey was completed by staff in one of the three animal care departments at Chimp Haven (husbandry, behavior, veterinary). We had multiple observers for each chimpanzee and ensured that each chimpanzee received a score from a member of each department, as the chimpanzees interact with each staff member differently depending on their relationships. Staff were asked to consider their observations of a chimpanzee during the prior two weeks as well as consider any abnormal behaviors that have been reported or observed. The number of wounds received during this timeframe, as well as their severity, should also be considered. The survey participant includes real-time observations as well as historical information on eating habits, locomotion, group dynamics, the chimpanzee’s ability to participate in normal activities as well as levels of staff concern regarding the individual (Appendix B). 

Wellness survey scores for each question are weighted differently based on the level of concern for a particular parameter. The scores from the survey participants are averaged by each parameter and then are added together to determine an overall wellness score for that chimpanzee at that particular time. Chimpanzees with higher cumulative scores require more intensive observation and may be placed on other types of behavioral metric protocols (Social Monitoring System or Targeted Individual Assessments) to determine negative impacts on welfare and possible intervention strategies. A cooperative feeding plan or other targeted positive reinforcement training plan may also be developed with the intent to decrease their scores and enhance their welfare. Lower scores indicate less concern and better welfare on this scale.

Total Wellness Score: The total score for an individual chimpanzee is the addition of all of the numerical scores of each behavioral category within the survey.Total = ABN + LOC + ACS + HAI + COT + SAG + SIA + SCO + ENR + WOUwhere the score variables are defined as follows:

ABN—Abnormal BehaviorSAG—Self-AggressionLOC—LocomotionSIA—Social Interaction ACS—Access to Food/WaterSCO—Staff ConcernHAI—Human Animal InteractionENR—Enrichment/Medication/Reinforcer AcceptanceCOT—Coat Appearance WOU—Wounds 

Scores can range from a low of zero to a high of 30. The scores are defined as follows: 

0–12—continue regular quarterly monitoring; 12–29—may require additional observations and interventions to decrease score; >30—report immediately to veterinarian and behaviorist; may begin quality-of-life monitoring. 

Wellness Surveys are ongoing, but for the purposes of this manuscript, we will concentrate on data from 16 June 2021 to 29 December 2022. Subjects were 298 chimpanzees (M = 123, F = 175, mean age = 32.34, range 4–61). A total of 24 observers (including behavior, husbandry, and veterinary staff members) conducted 1456 surveys on 303 chimpanzees during this time. The average wellness score for the colony during this timeframe was 1.52 (range = 0–10.33). All of the chimpanzees met the criteria to continue regular quarterly monitoring, and none met the criteria for additional observations or interventions based on the Wellness Surveys alone. The three biggest areas of concern for staff members were abnormal behaviors, coat condition, and wounding (Figure 2).

The Wellness Surveys allow for a collaborative view of each individual chimpanzee based on cross-departmental input. There has been anthro-zoological research which has suggested variations in observed non-human animal’s behavior related to differences in caretaker attitudes, handling practices, and caretaker familiarity [45,46,47,48]. The variation in assessor’s observations provides a more robust understanding of the overall calculated wellness of each individual chimpanzee. A large variation in survey parameter scores promotes opportunities for interdepartmental discussions to further investigate and better understand aspects of both the assessors and chimpanzees [49]. 

Although the survey is broad, it can be a helpful tool to document slower declines in welfare, especially in a species with a typical lifespan of 50 years or more [50]. By understanding cumulative data over time, it can supply vital information regarding patterns and trends during various life stages as well as general aging in chimpanzees. 

Originally, the survey was created on ZooMonitor, and survey assessors would access it there on a limited number of tablets which created a lack of accessibility for all staff. In August 2022, the decision was made to move the survey from ZooMonitor to Google Forms. In this way, the behavior team could email out the survey for assessors. Survey respondents from any department could then fill out the survey on any device of their choosing, thereby making it more accessible. The data are still exported in a similar way, and the survey questions remained the same in this transfer. The Google Forms’ survey format was also more commonly understood by the survey participants, which assisted with continued interdepartmental participation. 

## 4. Positive Reinforcement Training (PRT) Logs

Chimp Haven has had a Positive Reinforcement Training (PRT) Program since 2005. This program allows staff to progress through increasingly advanced levels to reach specific objectives related to training. The goals of utilizing PRT at Chimp Haven are to assist the staff with daily management procedures, including shifting chimpanzees from area to area, cooperative training for equitable feeding and enrichment opportunities, necessary veterinary procedures including training for body part presentation for wound and mobility assessments, medication compliance and advanced medical behaviors [51,52,53]. 

Records of each positive reinforcement training session are organized in Excel spreadsheets by chimpanzee social groups on a shared drive that is accessible to all staff. Each session is logged so it may be reviewed by other trainers who may train with that chimpanzee or their group and provide a quantifiable record of the session. PRT logs include information on requested behaviors, reinforcers used, the chimpanzee’s response to the cues or reinforcers, and the name of the staff member training the chimpanzee. It is noteworthy that we do not separate individuals from their groups for training sessions; staff engage with individuals while they remain in their social groups, and group mates may also be engaged with other staff at that time. A chimpanzee’s response to individual cues is measured on a three-point scale; a score of 1 designates that the chimpanzee seldom or never responded to the cue; a score of 2 designates that the chimpanzee needed assistance or further refining of their presentation of the behavior, or they were not consistent in their presentation of it; and a score of 3 designates the chimpanzee presented the correct response to the cue the majority of the time. 

Each group of chimpanzees also has a designated shifting log. The term shifting denotes the process of moving chimpanzees from one environmental area to another. This includes moving chimpanzees from inside their bedrooms to their outdoor enclosure or vice versa and moving between different enclosures. At Chimp Haven, shifting chimpanzees is a voluntary process that is typically reinforced with diet distribution. Interventions like utilizing PRT for stationing, enrichment placement, or providing access to novel areas may also be used to incentivize movement toward a desired location while also providing the chimpanzees with choice and control. The shifting log provides information on the staff member shifting the group, goal of the shift, the group’s specific location within the sanctuary, any special circumstances that may impact the ability to shift the chimps (e.g., weather), if all chimpanzees complied with the goal of the shift, and any notes that the shifter feels are noteworthy. These logs are completed each time the chimpanzees are shifted.

Data from the PRT logs are refreshed and pulled from Excel spreadsheets designated for each group of chimpanzees through Microsoft Power BI software (https://powerbi.microsoft.com/en-us/, accessed on 1 June 2023) and checked for errors. A session ID is generated for each unique training session utilizing the date, time, trainer, and chimp name. This creates data that are easy to filter to view values at the colony, group, or individual level. This review also includes monitoring, compiling, and sharing values such as colony engagement, trainer progress, and the total number of sessions with program stakeholders through regular reporting. The shifting log spreadsheets are reviewed for errors and values such as a group’s compliance with shifting goals and the ability of bedrooms to be routinely accessed on a biweekly basis. Trends related to shifting in varying temperatures, aggression from the chimpanzees toward shifters, or the efficacy of shifting plans are also monitored. 

PRT log data between January 2021 through December 2022 show that in 2021, staff trained with 301 chimpanzees (M = 122, F = 179, mean age = 32.3, range = 3 to 61) and in 2022, staff trained with 283 chimpanzees (M = 113, F = 170, mean age = 33.9, age range = 5 to 63). Subjects were limited to those who remained living during each year and those that did not arrive at the sanctuary that year. 

At the individual level, we focused on a 24-year-old female chimpanzee, Tessa. Tessa had low compliance when asked to move outdoors (shift) when her indoor areas needed to be accessed for routine husbandry procedures including cleaning and sanitation. To increase Tessa’s compliance with shifting, early training efforts focused on differential reinforcement of an alternative behavior. This included holding PVC tubes to assist with stationing or engaging her in a training session during shifting so doors could be manipulated by staff. She eventually progressed to more advanced behaviors unrelated to her shift training. For this case study, we reviewed Tessa’s PRT logs from January 2021 through December 2022.

Tessa was trained an average of five sessions each month in 2021 (58 total) and four sessions each month in 2022 (51 total). Tessa was trained extensively as a means of improving her shifting compliance. As training progressed with Tessa, it became necessary to vary the types of training and behaviors requested to keep her engaged. To this end, we progressed toward the following behaviors with Tessa: voluntary blood draw, injection, Kardia Mobile, Constructional Approach Training (CAT) [54], otoscope training, as well as temperature monitoring. CAT training involves the use of negative reinforcement, or removal of something undesirable to the trainee as a reinforcer, to reduce fear-based or aggressive behaviors. We applied CAT training to Tessa’s shifting plan to reinforce calm behavior through the removal of shifter attention and door manipulation, which was an aversive stimulus to Tessa as she had a history of reacting aggressively toward these aspects of shifting. This was also used in conjunction with positive reinforcement training to reduce the overall use of negative reinforcement. The logs aided in tracking the progression of Tessa’s CAT sessions as well as the efficacy of training during shifting, particularly in the notes section of her training logs. The notes contained information on Tessa’s behavior, how far the session progressed, whether or not doors were manipulated, behaviors requested, and Tessa’s reaction to door manipulations. This information was helpful for her trainer but also offered valuable tips to other care staff members about what worked successfully when shifting Tessa. 

Utilizing data from the PRT logs allows staff to communicate training trends with stakeholders such as facility directors, board members, and supervisors. For instance, from mid-August to mid-November 2021, our facility opted to exercise an abundance of caution and limit staff interaction with the chimpanzees, including most PRT training sessions due to an upper respiratory illness that impacted our colony’s health [55,56]. During this timeframe, the number of sessions declined from an average of 871 sessions a month from April through July of 2021 (four months prior to colony illness) to an average of 219 sessions a month from August through November. The impacts of staff turnover are evident during 2022, with reduced numbers of training sessions during this year compared to 2021. We also saw a large decrease from 946 sessions in March 2022 to only 162 sessions in September 2022. Monthly values for each trainer can also inform staff availability and management principles and assist in determining if our team members have enough time available to engage in PRT sessions. We are also able to track trainer engagement by department and chimpanzee. This metric is also useful for supervisors as they monitor and aid their teams in reaching their training objectives and performance goals. 

The PRT logs are tied to Microsoft Power BI; therefore, the logs can quickly provide a large quantity of information to PRT program staff and stakeholders. This format makes it quick to review trainee progress and staff involvement. Recently, PRT program staff were able to note that 100% of the colony had been recorded with at least one training session within the first few months of 2023. Without the logs being formatted in a way that allows them to be tied to Microsoft Power BI, this would have taken a significant amount of time to track and report. One current drawback of the training logs is that they do not provide opportunities for multiple trainers to be tied to the same session. This occurs when one staff member is mentoring another in the training program: only the new trainer’s name is listed when both the mentor and mentee attend a session. This is something PRT program staff hope to remedy in the future.

To demonstrate the utility of shifting logs and how they may inform decision making, we focused on Latoya, a 27-year-old high-ranking female chimpanzee residing in a group of six (M = 2, F = 4, mean age = 30 age range = 26–37) when we began collecting data in 2018. Latoya would position her body in the path of a closing door to avoid it shutting. Seeing the potential to use PRT to build trust and provide Latoya with agency over the shifting process, in 2018, staff implemented a shifting plan: a step-by-step training plan of how to ask for and reinforce chimpanzee movement [57]. Latoya’s shifting plan has been active and in use since that time. We examined Latoya’s shifting data from January 2018 through December 2021. Specifically, Latoya’s logs were reviewed monthly by her trainer, and values for compliance were determined by how many days she was asked to shift outside and complied, not taking into account days with weather conditions that prevented shifting or alternate goals of her husbandry team. Any trends were communicated with PRT program staff and Latoya’s care team to assess and include any modifications to her shifting plan. Latoya’s shifting compliance from 1 January 2018 through 31 December 2021 is displayed in Figure 3. Latoya’s shifting plan began in June 2018 with data from the shifting logs that catalyzed the creation of the plan. Latoya’s low shifting compliance was noted by multiple trainers in the logs which led to the creation of a plan that focused on consistency for Latoya. The plan also promoted time efficiency during shifting for staff, and it had the goal of increasing Latoya’s overall shifting compliance, which averaged 30% in 2018. Latoya’s compliance began to rise after the plan’s implementation, with 100% compliance in June, July, and August 2019. Latoya’s overall average compliance for 2019 was 74%. Latoya’s average percent compliance in 2020 was 70% and 49% in 2021.

Latoya’s shifting data were consulted to inform decision making and formulating her plan. Early in the plan’s implementation, from approximately June 2018 to mid-2019, Latoya was only shifted by her primary trainer and three other staff members rather than one dedicated individual [58]. Once additional staff members were trained on her plan, her average compliance dropped from its previous values. However, Latoya did maintain a relatively high percentage of compliance even throughout 2020 when she was shifted by new care staff members the majority of the time due to staffing constraints related to the COVID-19 pandemic. Had she regressed significantly, we would have reduced the number of new shifters working with her. In examining Latoya’s shifting data from 2021 specifically, it was also clear that her compliance changed dramatically in August and September of 2021 (0% and 20%, respectively) but seemed to improve some in October (68%). In August and September, trainers could not engage the chimps in training sessions to reduce the risk of spreading illness, and Latoya’s compliance with shifting decreased during that time from an average of 66% from April through July 2021 to an average of 35% from August through November 2021. This could be due, in part, to a reduction in interaction and additional training sessions outside of shifting. Both factors were integrated into her shifting plan early on as a means to build a relationship with Latoya [59]. In October, her shifting plan was updated to include stationing paired with reinforcement for desired behavior while doors were manipulated [60]. For Latoya, this meant remaining seated near the door and not standing, moving toward the door, or reaching for the door. When Latoya remained seated and engaged with the shifter, she received a continuous stream of juice while they manipulated the door. If her behavior changed, the shifter stopped reinforcing her until she demonstrated stationing behavior again. This update combined with increased interaction could have influenced the positive change in her compliance during that month. 

In contrast to the PRT logs, there is some room for improvement within the shifting logs. While they provide detailed records of every shift executed by staff, they take time to fill out and are not formatted in a way that makes useful information easy to discern. At present, the shifting logs contain variables that are not well defined. For instance, how is compliance defined? In the case of Bloomsmith et al. [61], compliance was measured by the percent of animals responding to the cue related to the shift, “Inside”. Compliance can vary between groups, shifters, and chimps. For many of our groups, compliance is looked at as following the goal of a shift. If chimpanzees are asked to shift out, compliance is all of the group shifting outside with any outliers noted as being noncompliant. However, for Tessa’s current shifting plan, compliance is defined as her exhibiting calm behavior (inactivity, eating/drinking, interacting with groupmates) and allowing doors to be manipulated safely but does not necessarily mean she shifts out with her group. This deviation from our typical definition of compliance was a result of her behavior when locked out, which included pacing and refusal of reinforcers. Defining the goal of the shift is also something staff hope to define in future logs. The most useful information from the logs is in the “Comments” section. This takes time to fill in and time for staff to read for relevant tips and information. This also involves the potential for human error, given that staff have to read through the various columns and comments to discern information. In future updates to these logs, the goal is to format them in a way that they are able to be tied to Microsoft PowerBI like the PRT logs. This would provide staff with data quickly and with little effort while still providing the necessary amount of detail.

## 5. Wound-Tracking System

One of the ways aggression is monitored is through the recording and quantification of the number and severity of wounds an individual chimpanzee receives. Chimp Haven’s veterinarians provide a caseload report of the wounds and their severity that chimpanzees within the colony have received during a given week. This information is then transcribed into a single master Excel Spreadsheet. The Microsoft Excel file is then used by Microsoft Power BI as a data source to illustrate the frequency and severity of wounding in individuals, groups, and the entire colony via graphs and tables that are shared with animal care team members. 

The Wound-Tracking System provides historical information on both the frequency and severity of wounds of individuals and groups. Veterinary staff record the healing progress of any wounds, while the Wound-Reporting System records only the initial occurrence of the wound or if the wound increases in severity grade. Wound severity is recorded using a five-point scale adapted from Ross et al. [62] (Table 1). This information assists with ongoing social monitoring of newly formed and existing social groups while also determining potential patterns of wounding in individuals or groups. 

The Wound-Tracking System uses these cumulative data to determine individuals of concern via our Chimpanzee Introduction Standard Operating Procedure wounding criteria (Table 2). If an individual or a group reaches the criteria in the Introduction SOP, the behaviorist, attending veterinarian, and the colony director discuss the group or individual at a weekly meeting to determine next steps. Group management practices such as cooperative feeding, additional forage, stationing for medication acceptance, PRT recommendations, and/or providing additional or novel space may then be implemented with the intent of reducing wounding frequency or severity. 

Even though these guidelines were created to follow the progress of chimpanzee introductions, these criteria pertain to all groups and individuals independent of the date of group formation. To determine chimpanzees of concern out of the data set, the Power BI dashboard has a page which then filters names of chimpanzees that match or exceed any of the criteria in the standard operating procedures. Another page then illustrates the wounding history over time of each of those individuals to provide additional context. Additional dashboards in the Power BI illustrate the wounding frequency and severity of each group and their individuals to help determine trends such as uneven distribution of wounding of individuals within a group.

Wounding data are collected on an ongoing basis and have been collected homogeneously from 1 January 2019 through 31 October 2022. However, we recently changed methodology and, in this review, focused on data from January 2022 to December 2022. Subjects were 309 chimpanzees residing at Chimp Haven during this timeframe (M = 125, F = 184, mean age = 32.76, age range = 4–62). 

In 2022, 56 chimpanzees were never reported to have a wound; therefore, only 253 chimpanzees are included in Figure 4, which shows the frequency and severity of wounds in Chimp Haven’s colony over time.

The majority of the wounds are superficial, and over half of the wounds were assigned a severity score of 1 (51.62%) while another 30.06% were assigned a severity score of 2 (Figure 5). Severity 3 wounds tend to be crushing wounds but can be more serious due to the difficulties inherent in determining the extent of damage in this type of wound (R. Jackson, personal communication). Severe wounds of scores 4 and 5 accounted for 4.71% of the wounds in 2022. 

To provide an example of group level wounding data, we focused on Pierre’s group (*n* = 17, M = 10, F = 7, mean age = 35.59, age range = 23–52) that underwent a series of three social integrations in November and December 2021 between two subgroups: Pierre’s original group (*n* = 13, M = 8, F = 5) and Sonya’s group (*n* = 4, M = 2, F = 2) consisting of individuals Sonya, Pamela, Kino and Duke. 

Pierre’s group experienced multiple wounds (*n* = 93) in the year following their introductions (Figure 6). However, most of these wounds (*n* = 68) were minor (severity scores of 1 and 2) and relatively evenly distributed among group members. Two males, the alpha, Pierre, and a lower-ranking male, Bryan, had no wounds reported during this timeframe; behavior staff have reported that both males are people-oriented and reluctant to engage in group aggression. Upon review of the wounding timeline, there are a few spikes in the levels of wounding in March, May, and October, which would correspond to three, five, and ten months following the introductions. We can also see that the introduced chimpanzees, particularly Pamela and Kino, were reported as having the most wounds during this timeframe, while Duke received the highest number of severity score 3 wounds. 

The Wound-Tracking System has proved to be one of the most useful metrics for monitoring the chimpanzees after social integrations. Although wounding levels in chimpanzees have been studied in chimpanzees housed in zoos and laboratory facilities in the past [62,63,64,65,66,67], current wounding levels for chimpanzees housed in modern sanctuaries which typically house larger groups and have larger spaces are limited [68,69]. The most recent article on wounding in chimpanzees looked at the effects of visitors on chimpanzee wounding in three zoos [70]. The Wound-Tracking System allows us to follow introduced groups and their wounding levels over time. For example, Pierre’s group has had lower levels of wounding in 2023 with lower severity scores than in 2022. This system also allows us to track individual chimpanzees and may alert us to chimpanzees engaging in wound picking of themselves or others. Once we are alerted to potentially high levels of wounding, we can begin various interventions that may assist in reducing the number of wounds. These interventions may range from separating individuals who have been introduced to providing additional forage or enrichment. 

Other benefits of the Wound-Tracking System include the ability to quantify the wounding levels of groups and even the entire colony. This enables Chimp Haven to create criteria for making various management decisions from moving a group of chimpanzees to a larger enclosure when wounding levels are increasing or providing additional distractions in the form of various enrichments. Wounding levels can be compared between groups of the same or different sizes, and updates on colony wounding levels can be sent to staff on a weekly basis to provide baseline information or denote chimpanzees of particular concern (Appendix C). 

Although the Wound-Tracking System is a supportive tool in the management of sanctuary chimpanzees, there are some drawbacks. Wounds may become infected or worsen due to self-picking which may increase the severity of a wound over time, thereby leading to changes in the overall severity scores per individual or group. Although staff are trained on how to grade the severity of wounds, sometimes, the wounds may be difficult to visualize, and scores may be more subjective unless a chimpanzee is anesthetized for treatment of the wound. This is particularly true for wounds with a severity score of 3, which may be more serious than at first thought. Chimpanzees, like humans, often have small superficial scrapes, which means that collecting data, particularly on the low severity score wounds, can be time consuming to gather and input. The biggest drawback to the Wound-Tracking System is that it does not include things like injuries to muscles or ligaments, mobility issues, broken bones that have not broken the skin, or other types of injuries that may not be as obvious in the same way as a wound. 

## 6. Hair Loss Surveys

Hair loss surveys track the chimpanzees’ body hair coverage to assess overall condition, as hair loss may be linked to increased levels of stress and abnormal behaviors, such as hair pulling [71,72]. Hair loss may also be the result of both temperature and humidity fluctuations, further increasing the importance of monitoring this potential physical indicator of wellness during different seasons. Underlying medical issues may also cause hair loss, and this may be an early indicator of some disease processes [73]. The behavior team performs hair loss observations on the whole colony once every 15 months. Chimpanzees new to the colony are also monitored when they first arrive for baseline levels of hair coverage. To better understand the entire behavioral profile of the residents of Chimp Haven, we also investigate potential causes of hair loss. 

Any chimpanzee that has lost 25% or more of their estimated cumulative total body hair is monitored at higher frequencies to identify the cause of the hair loss and suggest behavioral interventions if warranted. Three different behavior team members will observe each chimpanzee with 25% or more hair loss every four months. If a chimpanzee has an abnormal behavior related to hair plucking or hair eating, it is noted in their hair loss profile. 

We use ZooMonitor to record the estimated percentage of total body hair. A photo of a typical chimpanzee body, adapted from Ross et al. [62], is used with an overlay of a 100 × 100 grid. The chimpanzee body is divided into seven different sections, and each section has the estimated cumulative surface area score based on the number of grid segments residing within the chimpanzee body part (Figure 7). The estimated percentage of surface area per body part is determined by dividing the total number of grid segments counted within the pictured body by the total number of grid segments with that assigned pictured body part. Typically hairless portions of the body like the front of hands, bottom of feet, ears, and portions of the face are excluded. 

Observers are instructed to score the percentage of hair loss in one of the seven body zones. The observer estimates how much hair is missing from each body zone. Scoring is based on the following scale: 0% = no apparent hair loss, 25% = evidence of hair loss, 50% = significant hair loss in approximately half of the surface area of the zone, 75% = most of the surface area has hair loss, 100% = complete hair loss in that zone. To determine estimated total hair loss across the body, body zones scores of hair loss are added together with each body zone scaled to the estimated percentage of ideal hair surface area on a chimpanzee.

A total of 641 hair loss surveys were completed. From August 2021 to August 2022, 306 chimpanzees (M = 125, F = 181, mean age at start of project = 32.68, age range = 4–61) were surveyed and 189 chimpanzees were reported to have some degree of hair loss. Of the 306 chimpanzees observed, 117 were recorded as having 0% hair loss, 53 chimpanzees had hair loss of less than 5%, and another 53 had between 5 and 10% hair loss; 59 chimpanzees were recorded as having over 10% and less than 25% hair loss. Figure 8 shows the 24 chimpanzees with significant, or above 25%, hair loss during this timeframe. 

Surveys focused on individual chimpanzees, but we are also able to look at a group’s level of hair loss. We focused on Slim’s group, which was a group of 11 (M = 1, F = 10, mean age = 26.64, age range = 10–40).

Slim’s group had four individuals with significant hair loss and seven who did not (Figure 9). Group members who were surveyed and determined to have significant hair loss were then surveyed at a higher frequency than other group members per our hair loss survey protocol. The females in this group have close affiliative bonds and have a tendency to over-groom each other, often consuming the hair after plucking. We can address chimpanzees with significant hair loss attributed to abnormal behaviors (such as trichotillomania) by several methods, including increasing browse and forage, and by providing additional opportunities for species-typical or alternative behaviors [74]. From March 2022 through August 2022, additional forage was provided to Slim’s group in an attempt to curb hair plucking and over-grooming behaviors. The group was provided with routine forage three days a week; during this time, mixed nuts in the shell were provided on one of those days in place of their routine forage.

For the four chimpanzees in this group being surveyed for significant hair loss, their hair loss percentages decreased between March and July but increased again in November 2022. Angela’s percentage of hair loss decreased from 49.46% to 45.65% in June and down to 40.44% in July; however, in November, hair loss increased to 49.57%. Chakema had the highest levels of hair loss during our first surveys of this group at 72.19% in March but had reached an individual low of 54.05% by July; in November, she again had levels of hair loss above 70%. Nina’s hair loss remained relatively stable over time with consistent values in the 30% range. Slim also reduced her hair loss from 56.48% in March to a low of 45.94% in July with an increase back to 49.83% in November. 

Bischk, a 35-year-old female chimpanzee, was surveyed for significant hair loss (22.61%) in 2022. Her hair loss was particularly noticeable on her arms (Zones D and E), back (Zone C) and head (Zone A) (Figure 10). Upon analyzing Bischk’s hair loss over time, it decreased slightly from 25.8% in March 2022 to 19.9% in July 2022. During this time period, her group received mixed nuts in the shell as a supplemental forage. 

Hair loss surveys have become a beneficial tool in monitoring the change in hair loss over time for our chimpanzee colony. Coupled with other behavioral metric protocols, these surveys can help to determine possible causations of hair loss and measure if intervention strategies are effective. Staff prioritize the quick assessments of new arrivals to the sanctuary to create a baseline measure of hair loss that is then regularly monitored. This provides another layer of data which allows staff to monitor the chimpanzees’ adjustment to their new environment and social settings over time.

The first intervention strategy using hair loss data was with Slim’s group and Bischk. In both cases, it was decided by the animal care teams at Chimp Haven that providing additional foraging opportunities with mixed nuts could reduce hair plucking and overgrooming causing hair loss. The decision to provide additional forage was made based on Lutz et al. [74], where additional foraging opportunities reduced abnormal behaviors. During this time, we noted a reduction in hair loss in multiple chimpanzees during the time of additional forage intervention.

Although there were some general hair loss reductions over time in some individuals of Slim’s group, there were other variables that may have accounted for the change in hair loss. Between March and June, Slim’s group moved to different enclosures, which varied in outdoor and indoor dimensions and environmental complexity. Moving chimpanzees to large complex environments has been associated with positive behavioral changes that promote species-typical behaviors. The behavior changes include increased locomotive behaviors [75], prosocial behaviors [76], and expanded vegetation use for activities such as nest building and ant fishing [32,77,78,79]. A study by Ross et al. [76] found that when zoo chimpanzees moved from a non-naturalistic indoor enclosure to a more naturalistic indoor/outdoor enclosure, there was a reduction in abnormal behaviors. Captive chimpanzees moving from a larger complex space to a smaller less complex space also have been associated with behavioral changes such as aggression and self-directed behaviors [80]. During this period, Slim’s group was moved from a large open top outdoor enclosure with trees (January–mid-April) to a smaller less complex closed top enclosure (late April–May) and then back to a different large open top outdoor enclosure with natural vegetation (June–December 2022). It would be interesting to compare hair loss in different enclosure types in the future.

Another variable that may have contributed to overall levels of hair loss for Bischk and Slim’s group was the difference in temperatures and humidity between the seasons. For example, in one study on rhesus macaques, hair loss occurred most often in females during winter and spring [81]. Seasonal hair loss also occurs in vervet monkeys, aye-aye’s and lemurs [82] as well as other mammals [71]; for the vervets, hair loss is most notable from November through January. However, in humans, hair loss occurs most often, especially for women, in the hottest months of summer [83,84]. In the summer months, Chimp Haven, located in Northwest Louisiana, can have hot temperatures and high humidity. During the months the chimpanzees received the hair loss surveys, the average monthly temperatures varied from lows of 75.71 °F (24.28 °C) and 75.72 °F (24.29 °C) in November and March 2022 to highs of 89.54 °F (31.97 °C) and 92.76 °F (33.76 °C) in June and July respectively [85]. Dew points also varied from a low of 43.64 in March to a high of 72.23 in July. Humidity and other seasonal differences have influenced hair loss in macaques [81,86]. These differences in temperatures and humidity could potentially influence hair loss in the chimpanzees. For future studies over time, it will be interesting to compare and factor in these and other environmental changes when both determining and reviewing the effectiveness of intervention strategies.

One of the more difficult aspects of collecting hair loss data on individuals was the ability to have unobstructed views of different portions of each individual’s body. The survey assessors often would have to return to assigned chimpanzees over different points in time or would use positive reinforcement training to enable them to view various portions of the chimpanzees’ body. This process would often become time consuming and become difficult to accomplish in short timeframes for multiple chimpanzees.

Another difficulty that arose when training observers for this protocol was the inter-rater reliability in estimating the general percentage of hair loss on various segments on the body. The parameters of 0% or 100% hair loss had high levels of inter-rater agreement but parameters of 25%, 50%, and 75% hair loss took extra practice for assessors. Additional training with picture guides and in-person reviews were required for training new assessors to increase inter-rater reliability. Hair loss percentages were rarely exact, and the observer would need to estimate the general hair loss to one of the provided scores. 

For future studies, it would be interesting to study hair loss through image rendering technology. One idea would be to include three-dimensional imaging of chimpanzees moving through space and estimating surface areas of hair loss through imaging software. For example, Digital Life, a non-profit initiative within the University of Massachusetts at Amherst, creates high-quality and accurate 3D models of living organisms through various technologies of photography, scanning, and modeling [87]. This technology coupled with a focus on changes in surface area of hair could provide a more objective protocol to measure hair loss on individualized models for each chimpanzee’s unique body dimensions. In the event that this high-tech option is not available, taking a series of photos at many different angles and then using those images to estimate surface areas of hair loss may improve the reliability of the data and serve a similar function.

## 7. Targeted Individual Assessments

The behavior staff monitors abnormal behaviors through an abnormal behavior reporting system and follows through via Targeted Individual Assessments (TIAs). Anyone in animal care may report a chimpanzee who has displayed abnormal behavior. Reports typically come in via email and are transcribed into an Excel spreadsheet and monitored via Microsoft Power BI. Chimp Haven delineates different priorities for various abnormal behaviors based on their severity and potential harm to the chimpanzee. High-priority behaviors are ones that can potentially physically harm the chimpanzee (e.g., self-aggression/self-injurious behaviors, self-picking/wound picking), significantly alter their time budget in a negative fashion, and/or are newly occurring or previously documented that are increasing in frequency. Still important but of lower priority are behaviors that do not pose an immediate threat to the chimpanzee or their time budget but have the potential to do so or have recently increased in frequency (e.g., hair plucking/eating, rocking without injury, regurgitation and reingestion, pacing, and others). The lowest-priority behaviors include those that do not pose an immediate threat to the chimpanzee, do not alter their activity budget, and remain relatively infrequent. 

TIAs are triggered by the following: when higher-priority abnormal behaviors are reported, especially self-aggression and self-injurious behaviors; if severe or repeated wounding or notable group aggression is reported on the regularly monitored Wound-Tracking System; an abnormal behavior consisting of more than 10% of their activity budget is recorded by the Social Monitoring System; or notable behavioral changes such as a change in eating habits, unusual locomotion, or no longer engaging in species-typical behaviors are reported.

TIAs observations consist of two weeks of daily 15-min focal continuous observations on a chimpanzee of concern via ZooMonitor. After these data are collected and assessed, intervention strategies are developed to intercede with behavioral changes. Staff then follow up with an additional two weeks of observation either during or after any interventions, depending on the specifics of each case, to evaluate success or the need to try something different. TIAs generate an activity budget for an individual chimpanzee, which are then compared pre- and post-intervention(s) to determine any changes in the chimpanzee’s percentage of time spent engaged in various behaviors. 

We have completed 625 TIA observations on 27 individual chimpanzees collected through varying observation periods from March 2020 through October 2022 (M = 16; F = 11; mean age = 34, age range = 24–57). 

Because these assessments target individuals, to illustrate this type of assessment, we focus on a male chimpanzee named Kasey who arrived at Chimp Haven in 2006 and was 33 years old at the time of his assessment. When Kasey came to Chimp Haven, he was 18 years old and had a history of staph infections that impacted his right leg. Scar tissue had developed, and Kasey was managed with multiple medications over the years when wounded or in a flare. Medications included antibiotics as needed, antihistamines, non-steroidal anti-inflammatory medications, selective serotonin reuptake inhibitors, and medications for nerve pain. Kasey was the dominant male in his group from 2006 to 2019. In August and September 2019, Kasey’s group of 15 (M = 3, F = 12, mean age = 33, age range = 23–54) was introduced to another group that included three younger males (mean age = 25.67) and three females (mean age = 27.67). The new group was incompatible, and two males and one female from the introduced group were removed in December 2019 and were then integrated into another group. In 2020, Kasey resumed picking at his wounds, and in October 2020, he received a serious wound from a groupmate. In 2021, Kasey’s frequency of wound picking of both himself and others increased substantially from 13 total reports in 2020 to 29 reports between January and May 2021. Kasey was on TIA observations, and various intervention strategies were implemented from May to August 2021. These strategies included moving Kasey and his group to a larger habitat, increasing the frequency of daily forage from one to three times per day, adding laser therapy and CBD oil, altering how the group was shifted, and increasing Kasey’s engagement in positive reinforcement training. 

Kasey’s level of wound picking decreased between May and August 2021 (Figure 11). The first intervention attempted was providing a forage mix to the group three times a day (instead of just one time daily); this began in early 2021, and we did not see a decrease in Kasey’s wound picking. The next three interventions were instituted at the same time. Kasey’s group was moved from an open-air corral to a larger forested habitat in May 2021. In addition, group-shifting procedures were altered to ensure that they were not secured in a particular area for an extended time. Laser therapy on Kasey’s wounds and the provision of CBD oil also began at this time. In July, we began increasing the number of PRT sessions with Kasey and included the use of sensory targets for Kasey to interact with to distract him from picking his wounds; Kasey’s voluntary participation in these sessions was recorded as “engage”, while “intervention” included the provision of forage items. It is important to note that although Kasey’s engagement with training was highest in July (6.81%) and that was the month with the most sessions (17), the high level of engagement may have been a result of PRT sessions occurring at the same time as observations during that month. In July, Kasey’s wound-picking behavior was reduced to 0.75% of his behavioral time budget from a high of 5.71% in May; Kasey’s wound picking decreased to 0.26% in August 2021. 

TIAs were one of the first behavioral metrics at Chimp Haven. These assessments were instated to assist in providing objective behavioral data on individuals of concern for abnormal behavior, difficulties navigating the social relationships in their groups, or mobility concerns. These assessments are not collected through instantaneous sampling or surveys but provide more specific information on individual chimpanzees via observations on a single focal animal [44]. These protocols are flexible and can be adapted in terms of the duration, frequency and timing of observations based on the concerns for a specific chimpanzee. Abnormal behavior can be very specific to an individual, and interventions and observations may also need to be individualized for optimum effectiveness [88]. 

In Kasey’s assessment and follow up, multiple interventions were attempted, and any of these variables may have singularly contributed to Kasey’s decrease in wound picking, or it may have been the combination of interventions. Medications typically reduce abnormal behavior in non-human primates but may have significant side effects and may only work for the timeframe they are given [89]. Often, medications are a last resort for social species due to their potential to sedate the animals and for great apes in particular, concerns for side effects that impact the heart [R. Jackson, personal communication]. With such extreme abnormal behavior, it became difficult to control potential influencing variables as interventions were quickly added to ensure Kasey’s well-being. A combination of therapies including changes in husbandry, medications, and positive reinforcement training is recommended for the treatment of non-human primate abnormal behavior [89]. In a more controlled setting with a less serious abnormal behavior, the TIAs observations may have more readily determined what specific intervention was most effective. Kasey’s wound picking was so severe that waiting to determine which individual variable was effective was not an option. Even if a single intervention strategy was not determined to be effective, a review of all the events and strategies used can be replicated for future chimpanzees with similar abnormal behaviors, and perhaps this particular combination of interventions could mitigate those abnormal behaviors as well.

## 8. Social Monitoring System

The goal of the Social Monitoring System (SMS) is to determine behavioral activity budgets and social relationships for all of the chimpanzees at Chimp Haven. Each chimpanzee is observed at least six times per year for 10-min focal follows, which include instantaneous and all-occurrence scan sampling [45]. SMS utilizes an ethogram of 34 instantaneous behaviors, six all-occurrence behaviors, and records the proximity of neighbor(s) within one meter of the focal chimpanzee (Appendix A). Observations are recorded via ZooMonitor. Significant events such as the passing of a group member, social introductions, severe or accumulated wounds, new arrivals, and releases into novel areas may trigger additional observational periods. Observation periods vary depending on the reason for the observations (Table 3). SMS provides the percentage of time each individual, group, or the colony as a whole spends in particular behavioral categories to provide us with behavioral time budgets at each level. This allows for comparisons between individuals, groups, or the colony overall. 

Observers are randomly assigned times and focal animals and observe between the following timeframes: 07:30–10:30, 10:30–12:30, and 13:30–15:30. For this manuscript, we focused on colony-wide data from 1 May 2018 to 31 October 2022 encompassing 317 chimpanzees (M = 128, F = 188, mean age = 29.37, age range = 1–58). 

The Social Monitoring System provides the broadest look at the behavioral time budgets of Chimp Haven’s chimpanzee population over time (Figure 12). During the timeframe of 1 May 2013 to 31 October 2022, staff collected 14,622 observations on the chimpanzees. The chimpanzees spent 51.17% of their time inactive during this period. The next most frequent behavior was food/drink or appetitive behaviors at 16.24%. The chimpanzees at Chimp Haven are fed produce and commercial biscuits twice a day. In addition, they receive enrichment foods and forage mixtures regularly; chimpanzees in the habitats can also forage for natural vegetation. Self-directed behavior includes self-grooming and is the next most common behavior at 10.57%. Locomotion and affiliative behavior accounted for 9.07% and 6.68% of their time budgets, respectively. Both abnormal and aggressive behaviors accounted for less than 2% of the time budget. 

Because one of the initiators for beginning social monitoring is a social introduction, we reviewed the data on two recently introduced groups. Blue’s group (M = 3, F = 3, mean age = 34.67, age range = 32–38) was introduced to Rosa’s group (*n* = 5, M = 1, F = 5, mean age = 29.8, age range = 26–38) in March 2022 to create a group of 11. 

We can also look at the time budget of an individual within a group and analyze how their behavior compares to that of other members in their group. We focused on Blue, a 37-year-old male chimpanzee that arrived in February 2021. We compared Blue’s time budget to that of his group mates from March 2022 to February 2023. Blue’s group mirrored the colony data with inactive, food/drink, self, affiliative and locomote making up the greatest percentages of the group’s time budget (Figure 13). Blue, however, differed in some ways. Blue was less inactive (31.43%), less affiliative (5%), and spent a greater percentage of his time in almost all other behaviors including abnormal behavior (3.57%) except for aggression and sexual behaviors. 

The Social Monitoring System provides Chimp Haven with an overall time budget for our specific chimpanzee colony as well as for particular groups or individuals. Although we can, and do, compare behavior in a variety of settings including zoos, sanctuaries, research facilities, and even the wild, each place is unique [90]. Environments for chimpanzees vary; chimpanzees live all over the world in various settings with different types of enclosures, different diets, and even different standards of care [90,91,92]. More specifically, the same can be said for zoo, research, and sanctuary chimpanzees in North America. Each chimpanzee colony is distinct. Therefore, being able to compare our colony’s activity budget to itself over time, during different seasons, under different social conditions, and with the chimpanzees in various types of enclosures provides objective data specifically for the chimpanzees residing at Chimp Haven. 

The Social Monitoring System activity budgets can be compared over time to look for changes related to specific events: for example, a move to a new enclosure type. Chimp Haven has multiple types of enclosures to provide individualized care, but the chimpanzees can also be rotated into different areas to encourage an exploration of novel spaces. We can also compare groups of similar ages, group sizes or even compare groups undergoing different stressors like new neighbors, an introduction to new group mates or construction in the area.

There are some limitations to this system. When the Social Monitoring System was initially piloted, the goal was to collect both activity budget data and social information with a limited number of observers and limited observation time. In order to capture directional social behaviors, the system uses all-occurrence behavior and social proximity information but does not include information on specific social behaviors like approaches, displacement, or other social behaviors that occur between specific individuals. In the future, it would be interesting to add grooming relationships and social dominance hierarchy rankings to this system given the many studies that have reported on the various effects of rank and relationship quality on chimpanzee behavior [93,94,95,96].

## 9. Nearest Night Neighbor

The purpose of the Nearest Night Neighbor protocol is to determine the preferred locations and social partners of individual chimpanzees during the nighttime hours. This protocol focuses on individuals in groups that were recently introduced to new chimpanzees, chimpanzees that may be introduced in the near future, and newly arrived chimpanzees. 

Chimp Haven employs night-time caregivers to provide regular nightly monitoring of the chimpanzees. Each of these team members records whether an individual chimpanzee is located inside or outside, the nearest neighbor’s proximity (Table 4), and the nearest neighbor’s name. The night-time caregiver records the information via ZooMonitor in a single instantaneous sampling interval between the hours of 19:00 and 00:00. To approximate the strength of social relationships within a group, social network sociograms are created from these data. The sociograms are analyzed and modeled through R from the exported file from ZooMonitor. The sociogram calculates the frequency of nearest neighbor pairings through every member of the group. The thickness of the line between two chimpanzees indicates the number of nights they were recorded as nearest neighbors. Thicker lines indicate greater frequencies of pairs being nearest neighbors. In addition, the location of a chimpanzee’s marker and its distance from another chimpanzee identifies the strengths of the bonds between them. 

Chimpanzees at Chimp Haven have the opportunity to choose their Nearest Night Neighbor as well as selecting the location where they rest/sleep throughout the night. The Nearest Night Neighbor data also record the chimpanzee’s observed location, either indoors or outdoors. Chimp Haven allows indoor and outdoor access for all chimpanzees in the colony at all times, unless for special circumstances like extreme weather where access may be restricted or during the cleaning and sanitization of an enclosure. This portion of the data is monitored as differences in the location preferences of individuals may show key indications of progress of social bonding or the lack thereof.

To illustrate the Nearest Night Neighbor data collection protocol, we focused on Blue’s initial group (*n* = 6, M = 3, F = 3, mean age = 34.67, age range = 32–38). To form this group, three pairs of chimpanzees were introduced to each other in March 2021. 

Data were collected on the proximities of Blue’s group for 92 non-consecutive days between 4 March 2021 and 30 June 2021. An overall sociogram was developed which includes all proximities (Figure 14). Alexis and Isabelle resided close to each other with a high frequency of being nearest neighbors at night, which is indicated by a thicker line (Figure 14). Others, like Blue, resided on the periphery of the model, indicating low frequencies of being or having a nearest neighbor, which is indicated by thinner lines. Mitch was nearly central between Candi, Cecil, Isabelle, and Alexis, which indicates a more even distribution of being a nearest neighbor to these individuals. 

Figure 15 provides additional context by reviewing each additional proximity parameter to help determine potential social significance. For example, a chimpanzee that is often in contact with another chimpanzee as their nearest neighbor may be more socially significant than having that nearest neighbor be within three meters. 

Blue did not have a neighbor in his space for 39 out of the 92 nights, while Isabelle was without a neighbor in her space for only one night (Figure 16).

Blue’s group spent all 59 nights that they were observed indoors between the months of March through May. In June, four individuals, Blue, Cecil, Candi, and Mitch, spent one night outdoors and 19 nights indoors. 

The Nearest Night Neighbor data provide a first look at social relationships when chimpanzees first arrive at the sanctuary in existing social groups and when they are newly introduced. Research in non-human great apes has suggested that social proximity may play an important role in conflict resolution and may predict which individuals are likely to be affiliative with one another [97,98]. This information provides us with a better understanding of individual and group dynamics. This can assist in further decisions regarding social integrations by suggesting where strong bonds exist and who might (or might not) support them as they meet new groupmates. For example, this information is routinely accessed and shared when determining the order of adding chimpanzees, particularly in large group introductions. This helps Chimp Haven better estimate existing social bonds to ensure that both new arrivals and individual chimpanzees have social support during an introduction. 

Location information is particularly useful when a group or individual is taking longer to integrate. Individuals or subgroups may remain outside when the rest of the group routinely remains inside during the night. In inclement weather or extreme temperatures, it becomes even more important to know if a chimpanzee is remaining outside overnight. It is important to allow individual chimpanzees sufficient time to acclimate to new social pairings, but intervention methods may be implemented for chimpanzees that remain in outdoor enclosures during temperature or weather extremes. Intervention strategies may include providing additional heating or cooling options, outdoor shelters, or possibly additional indoor locations and increased space. 

Although the Nearest Night Neighbor data do provide us with some information on proximity and location, we recognize that the data are limited. The observations only take place once each night, while the chimpanzees may change their location and nearest neighbor many times throughout the night. Unlike other Social Network Analysis (SNA), Nearest Night Neighbor does not include social behaviors that would supply additional information and context regarding social bonds and hierarchies [99]. Therefore, Nearest Night Neighbor data are most effective when used in tandem with other behavioral metrics. 

## 10. Discussion

It has become common practice when discussing chimpanzee behavior to compare the activity budgets of wild and captive chimpanzees [100,101]. This practice, although informative, may not provide an accurate assessment of captive chimpanzee welfare [102]. Instead, it may be more beneficial to compare captive chimpanzees to other captive chimpanzees; however, this may also have drawbacks due to differences in captive environments that may affect behavior [12,103]. Facilities that house captive chimpanzees also vary greatly in their construction, provision and type of space, as well as philosophies of care, even among sanctuaries. What may be most beneficial and provide us with the best indicators of chimpanzee welfare may be to compare chimpanzees at the same facility to one another and look at changes in an individual chimpanzee’s behavior over time. Creating activity budgets, as well as collecting other metrics for individuals and groups or a particular colony, may provide us with the most accurate baseline information, comparisons, and data on effective interventions. For example, another chimpanzee sanctuary, Save the Chimps, in Florida, U.S.A. creates individualized care plans to determine behavioral baselines for their residents [104]. These baseline data have allowed staff to create custom plans for individuals which have resulted in decreased abnormal behaviors and increases in social interactions and enrichment use. 

Chimp Haven has built and piloted a number of behavioral metric protocols over the past five years to provide staff with an objective and comprehensive picture of the chimpanzees’ behavioral welfare. The information on the chimpanzee’s behavior based on these protocols is shared with Chimp Haven staff members regularly, including via weekly emails to all animal care staff regarding the Wound-Tracking System, monthly updates to supervisors and directors regarding positive reinforcement training progress, and easy-to-understand graphs on a shared drive for easy access. 

One of the goals of using the collective behavioral metric protocols is to help answer one of the most common animal management questions: how is this individual or group doing? The collective information helps to guide managers in creating informed hypotheses, determining areas of concern, and identifying potential methods to intervene or mitigate undesirable behaviors. The layering of all of the behavioral metrics complements the other metrics collected from other animal care departments at Chimp Haven to provide a well-rounded view of the overall well-being of both groups and individuals. Since this information is shared throughout the facility, it allows for efficiencies in interdepartmental communication, cooperation, and problem solving when managing a large colony of over 300 chimpanzees.

In specific cases of concern, all of the behavioral metric protocol information can be presented in an Integrated Welfare Report. This can focus on various questions and concerns regarding the overall colony, group, or an individual chimpanzee. This report provides staff with additional information to provide a more robust behavioral narrative with key takeaways, objective behavioral trends, and the ability to compare with other groups or individuals. This report assists in managing the behavioral welfare of individuals and groups and provides us with an objective review of various aspects of their daily lives. The report allows animal care staff to easily cross-reference each protocol to determine relationships and better develop intervention strategies for individual chimpanzees or groups of concern. The efficiencies that the reports provide are most important in the behavioral management of events like social integrations or bouts of severe abnormal behavior. In these cases, rapid, educated decisions may need to be made regarding possible intervention strategies and determining the effectiveness of those interventions. Chimpanzee caregivers, equipped with this myriad of behavioral data, may be better equipped to quickly assess and objectively problem solve. The Integrated Welfare Report provides a method to quickly and effectively monitor individuals and groups, particularly before, during and after various experiences in the chimpanzee’s lives such as introductions or following the death of a groupmate. 

The various types of data collection on the chimpanzees’ behavior discussed here consist of both proactive and reactive methods to assist us in determining an individual chimpanzee’s overall behavioral welfare. Each of these behavioral metrics provides us with additional information about the chimpanzees’ overall welfare. The use of technology allows us to view and compare behavior, wounding, and hair loss levels over time. However, technology can have drawbacks such as being inaccessible during times of power outages or system failures. Another issue that has been noted in our system is the misspelling of chimpanzee names between observers, which may lead to confusion or incorrect frequencies or counts of behaviors. This issue can be surmounted by creating a fail-safe mechanism in the program to catch misspelled names and alerting staff to assign the correct name. 

Determining the causes of hair loss, abnormal behavior, or deviations in activity budgets can also be complex with the potential for multiple causes contributing to these negative indicators of welfare. The problem of multicausality also exists as we attempt to determine if a particular intervention has been effective. In many cases, multiple interventions are instituted at the same time in an effort to quickly reduce any negative behaviors and enhance the welfare of the individual chimpanzee. 

Welfare monitoring and assessment can take many forms depending on the species of concern and their natural history, needs, and their environment, i.e., captive welfare assessments may be very different from assessments of chimpanzees or other species in the wild [102]. There have always been and will continue to be various ways of both defining and assessing welfare [105,106,107,108,109,110]. In addition, new ways of measuring welfare, particularly with new technologies, are continuing to be investigated and discovered [111,112]. For example, recent studies have looked at various aspects of locomotion as an indicator of welfare for chimpanzees [40,113,114], and another recent study looked at the effects of sleep disruption on chimpanzee well-being [115]. 

## 11. Conclusions

Each type of data collection provides objective information on various aspects of the chimpanzee’s behavioral well-being at Chimp Haven. Although there are benefits and drawbacks to each method, over the past five years, Chimp Haven has begun to hone what works for our facility and staff. We have been fortunate to have a large, dedicated animal care teams who can provide input on, and perform observations of, the chimpanzees in our care; however, we feel that many of these types of assessments could be easily and readily adapted and utilized by smaller teams for the individuals in their care. Assessing welfare continues to be a growing and evolving science, and the authors look forward to determining how best to continue assessing and improving the welfare of chimpanzees and other animals in the future. 

## Figures and Tables

**Figure 1 animals-13-02595-f001:**
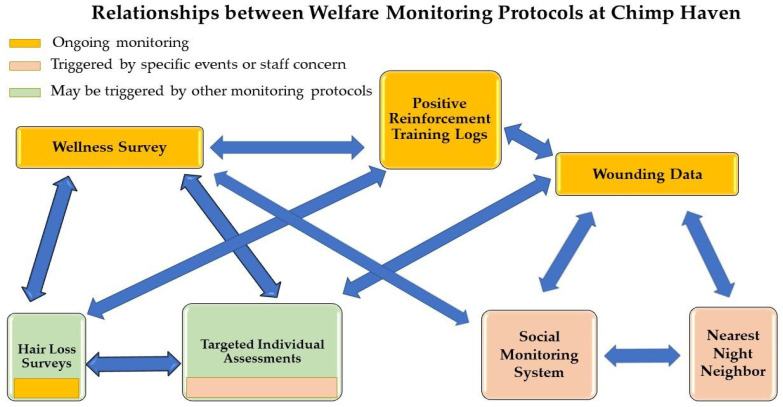
The relationships between the seven methods of objective and quantitative behavioral monitoring at Chimp Haven.

**Figure 2 animals-13-02595-f002:**
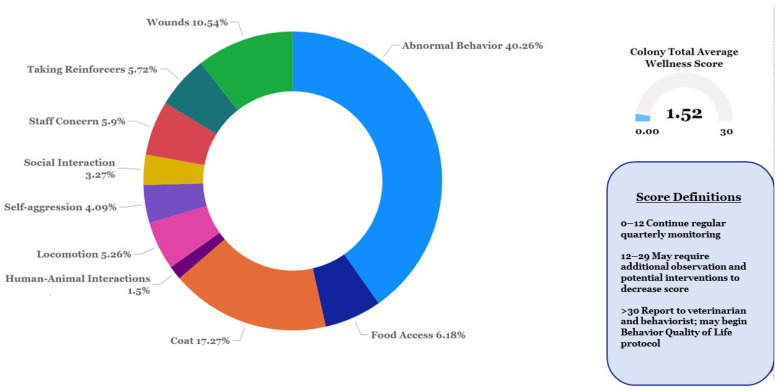
Percentage of chimpanzees of concern in each category of the Wellness Survey from 16 June 2021 to 29 December 2022.

**Figure 3 animals-13-02595-f003:**
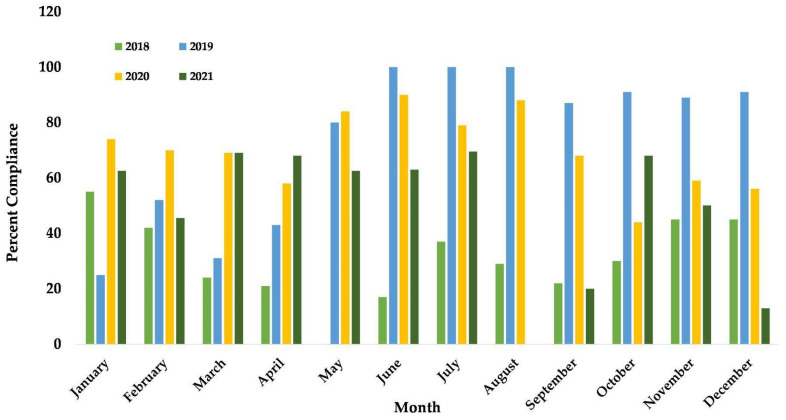
Percent compliance for Latoya shifting outside from 2018 to 2021.

**Figure 4 animals-13-02595-f004:**
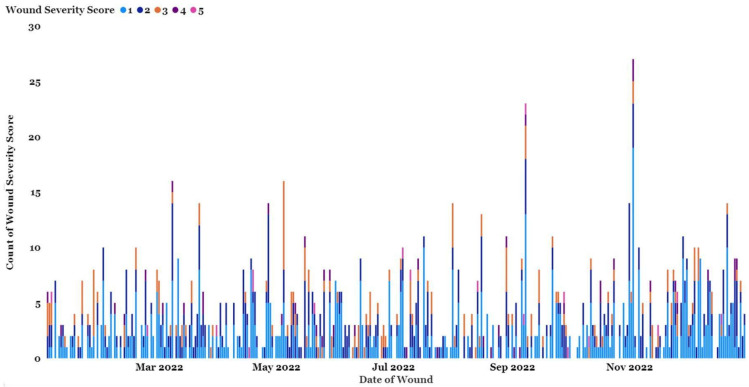
Frequency and severity of wounds per day during 2022 for 253 chimpanzees.

**Figure 5 animals-13-02595-f005:**
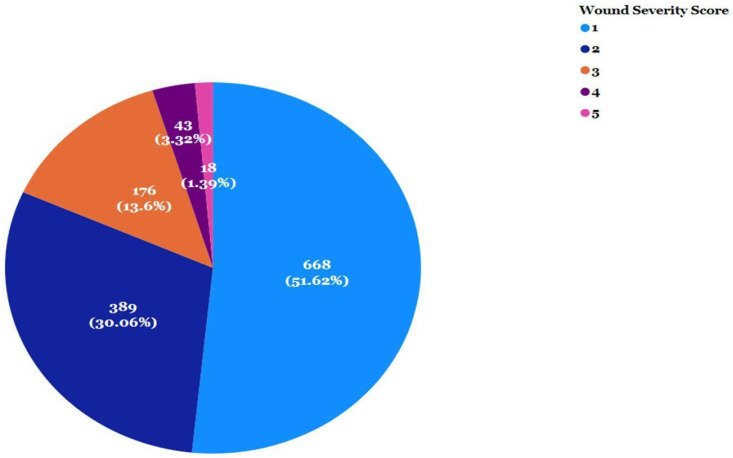
Wound severity scores for 253 chimpanzees for 2022. Wound severity scores are defined as follows: 1 = Superficial, scratch/scrape, partial skin break; 2 = Shallow cut, full skin break; 3 = Moderate wound, less than one inch deep; 4 = Deep wound, more than one inch deep; 5 = Severe wound, gaping, or missing body part.

**Figure 6 animals-13-02595-f006:**
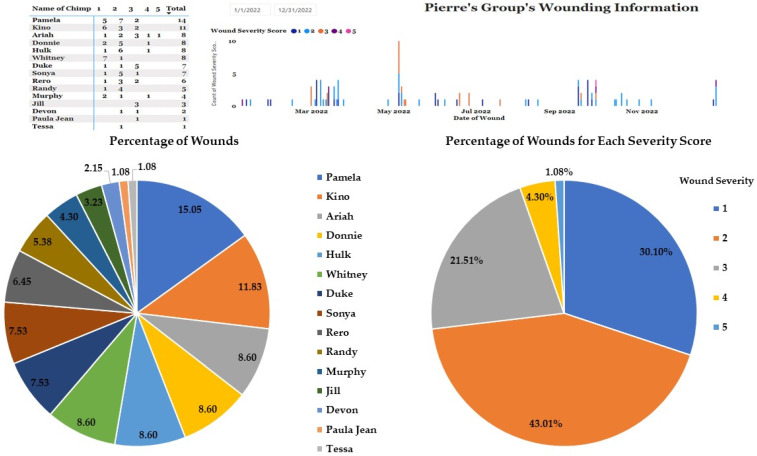
Wounding frequency and severity for Pierre’s group in 2022 following social integrations in December 2021. Two group members were not reported to have any wounds during this timeframe.

**Figure 7 animals-13-02595-f007:**
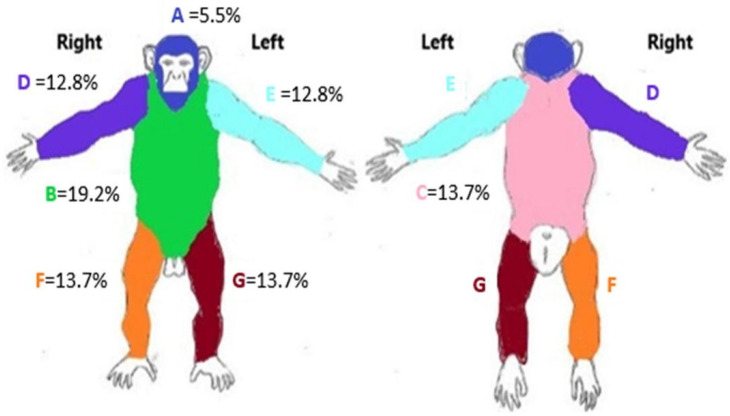
Hair loss survey body regions and percent of total body surface area in each zone diagram. Colors and letters correspond to one of the seven different body zones of the chimpanzee.

**Figure 8 animals-13-02595-f008:**
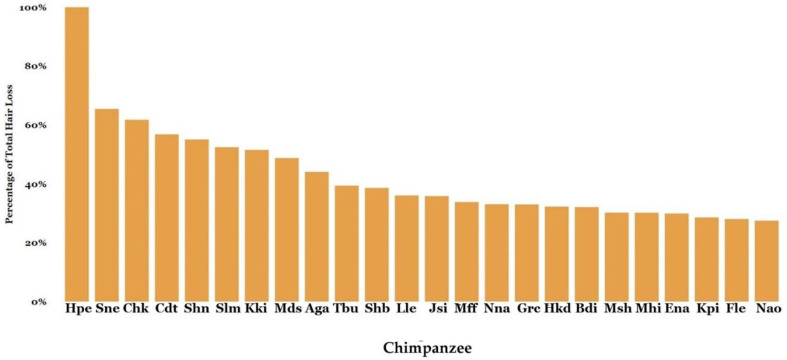
Percentage of total hair loss for chimpanzees with significant hair loss from 6 August 2021 to 2 August 2022. The top two chimpanzees both have a medical condition, *alopecia totalis*.

**Figure 9 animals-13-02595-f009:**
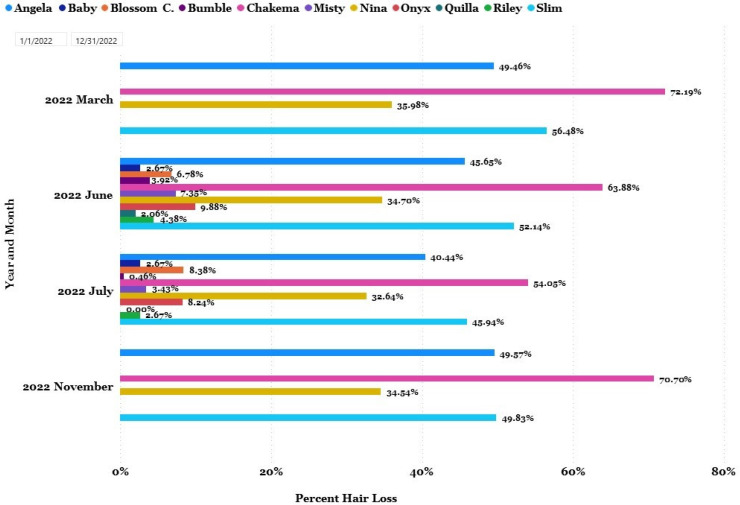
Hair loss totals for the individuals in Slim’s group. In March and November 2022, only the chimpanzees with significant hair loss (Angela, Chakema, Nina and Slim) were surveyed.

**Figure 10 animals-13-02595-f010:**
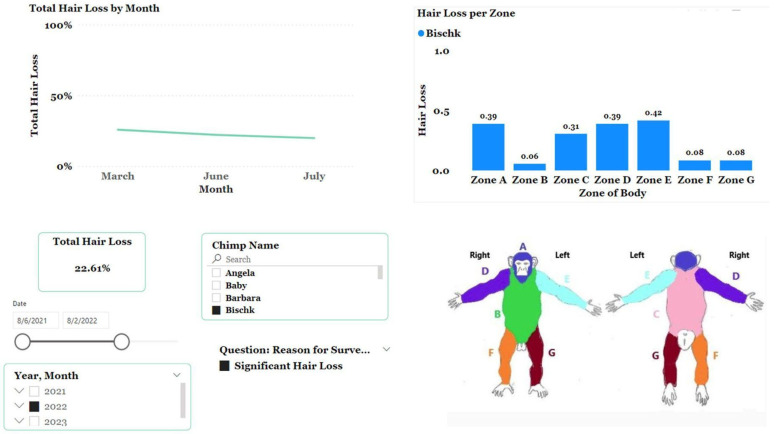
Hair loss survey results for Bischk for 2022, including hair loss by month and per zone. Letters depicted on the chimpanzee body illustrations correspond to the zones listed above them in the bar graph.

**Figure 11 animals-13-02595-f011:**
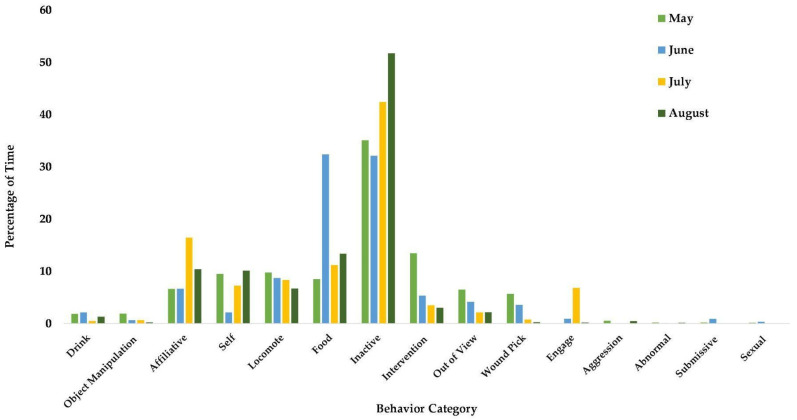
Kasey’s Targeted Individual Assessment comparison of various behaviors over time and with various interventions while in the habitat at Chimp Haven. Positive reinforcement training was recorded as “engage”. “Intervention” was the provision of forage. Kasey was observed during 15-min focal observations but session numbers varied/month; May = 9, June = 14, July = 22, August = 13.

**Figure 12 animals-13-02595-f012:**
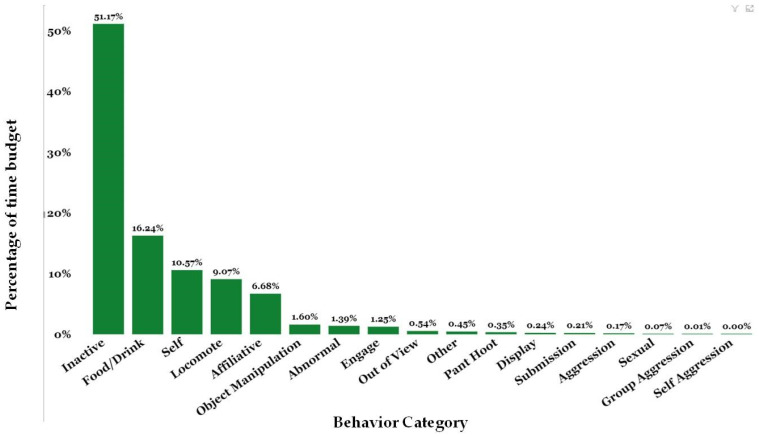
Percentage of time spent in various behavioral categories for 317 chimpanzees at Chimp Haven from 1 May 2018 to 31 October 2022. Data collected via Chimp Haven’s Social Monitoring System.

**Figure 13 animals-13-02595-f013:**
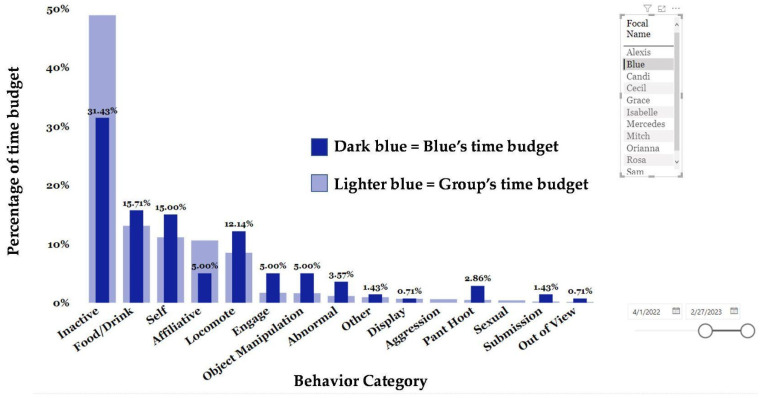
Blue and his group’s percentage of time spent in various behavioral categories from 1 April 2022 to 22 February 2023. Data collected via Chimp Haven’s Social Monitoring System.

**Figure 14 animals-13-02595-f014:**
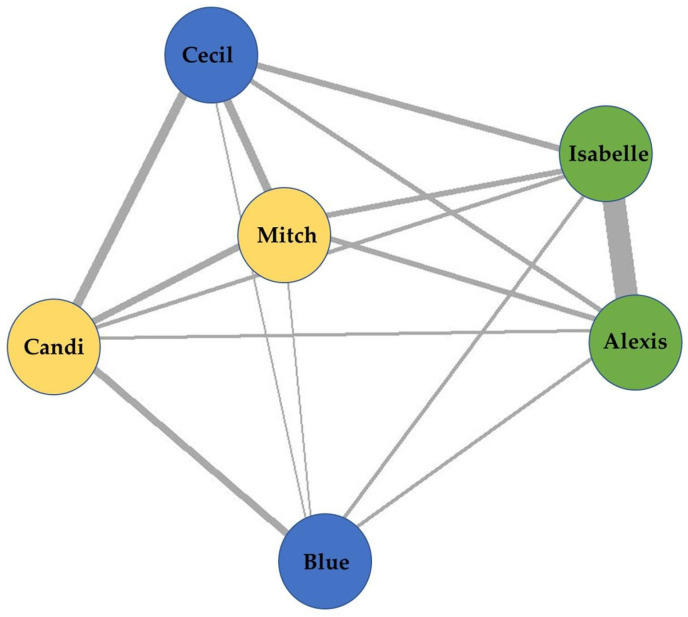
Social network sociogram developed from Nearest Night Neighbor data on Blue’s group. This sociogram includes all proximities. Each color indicates the original subgroups that were introduced to this larger group.

**Figure 15 animals-13-02595-f015:**
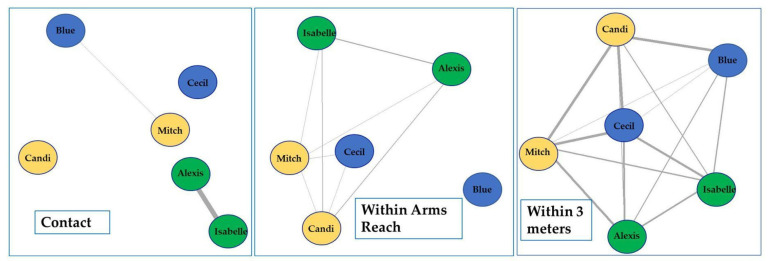
Social network matrices developed from Nearest Night Neighbor data on Blue’s group including the following proximities: contact, within arm’s reach and within 3 m.

**Figure 16 animals-13-02595-f016:**
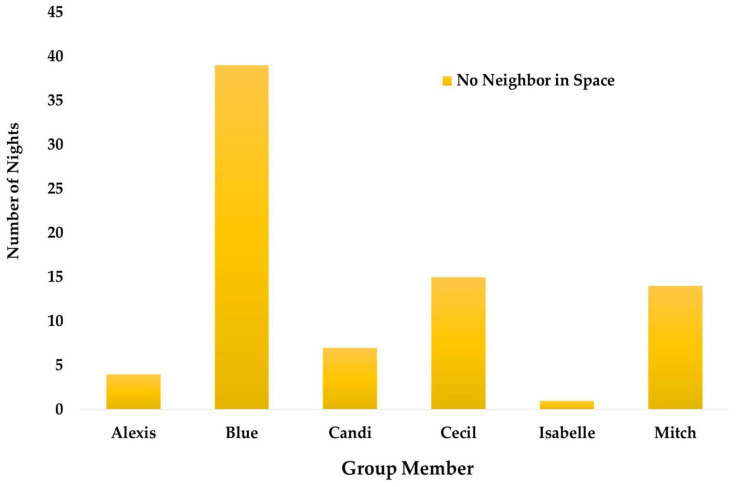
Number of nights each group member had no neighbor in their space during Nearest Night Neighbor data collection on Blue’s group.

**Table 1 animals-13-02595-t001:** Definitions of wound severity grades for input into the Wound-Tracking System.

Severity Grade	Definition
1	Superficial, scratch/scrape, partial skin break
2	Shallow cut, full skin break
3	Moderate wound, less than one inch deep
4	Deep wound, more than one inch deep
5	Severe wound, gaping, missing body part

**Table 2 animals-13-02595-t002:** Specific criteria on the number of wounds and their severity reported for an individual in a two-week period that meet our criteria for a higher level of concern. These criteria are included in Chimp Haven’s standard operating procedure on introductions.

Wound Severity Score	Number of Wounds in a 14-Day Period to Meet Criteria of Concern
A total of 10 wounds of any grade	
3	4
4	3
5	1

**Table 3 animals-13-02595-t003:** Social Monitoring System observation duration and frequency depending on reason for observations.

Reason for SMS Observation	Length of Time to Complete	Number of Observations per Chimpanzee
Post-Social Integration	3 months	13
Separation from Group	3 months	13
Yearly Check-In	6 weeks	6
Wounding/Aggression/Welfare	6 weeks	6
Considered for Integration	6 weeks	6
Release to Novel Area	4 weeks	4
New Arrival	3 months	13
Pre-Social Integration	6 weeks	6
Wounding/Aggression Check-In	6 weeks	6
New Arrival 6 Month Check In	4 weeks	4
Integration Check In 6 Months	4 weeks	4
Integration Check-In 1 Year	4 weeks	4
Death of a Group Member	6 weeks	6

**Table 4 animals-13-02595-t004:** Nearest night neighbor proximity definitions.

Proximity	Definition
Contact	The focal chimpanzee is in physical contact with another chimpanzee.
Within Arm’s Reach	The chimpanzee is within arm’s reach of another chimpanzee.
Within 3 Meters	Another chimpanzee is more than an arm’s reach away from the focal chimpanzee but less than 3 m away.
Outside of 3 Meters	The focal chimpanzee is more than 3 m away from another chimpanzee. May be used when the nearest neighbor is in another room.
No Neighbor in Shared Space	The focal chimpanzee has no other chimpanzee in the same room or external area. The chimpanzee is alone in whatever space they occupy.

## Data Availability

The data presented in this study are available on request from the corresponding author. The data are not publicly available due to restrictions regarding data sharing.

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
