# Peer review of "Behavioral Welfare Research for the Management of Sanctuary Chimpanzees (Pan troglodytes)"

_animals, 2023, doi:10.3390/ani13162595_

Round 1

Reviewer 1 Report

This study introduces seven behavioral metric protocols constructed and piloted at a large chimpanzee sanctuary in the US (Chimp Haven). Each behavioral metric is described in detail alongside an example of the corresponding data produced. There is then further discussion of how the data are interpreted and how decisions relative to chimpanzee welfare can thus be made. Improvements and self-critiques are also offered, which would prove useful to a reader who is looking to implement similar data collection methods within their own setting. The metrics themselves use a variety of data collection methods (observations, survey), programs (Power Bi, Google Forms, Excel), require input from a diversity of staff members (behavior, husbandry, veterinary), and operate at varying levels of complexity (completing a subjective survey to more time-intensive PRT training to social network analysis). The case studies provided for chimpanzees at Chimp Haven for each metric are helpful supporting illustrations of these metrics in action. 

This is important work for the larger caregiving and sanctuary community, not only because there is a lack of sanctuary-specific publications within the literature compared to other settings (i.e., zoo-housed or field studies) but also because behavioral resolution, as demonstrated here, can be specified to the individual, group, or colony level, and it can be done so longitudinally. Added benefits include the authors’ addressing of metric limitations as well as suggestions for improvement or expansion. All of this information will be valuable to a reader interested in improving their own methodologies or data collection schedules. Although the sanctuary community doesn’t necessarily agree whether research should be performed on the inhabitants, there is significant value in initiating a transparent dialogue about data collection methods within the sanctuary setting and how that data can be assessed to make choices that positively impact the welfare of the residents.

My suggestions for revision are minor and mostly revolve around consistency throughout the manuscript for formatting, spacing, and some mechanics. This should be relatively easy adjustments for the authors to make.

Overall:

·      Check spacing at the start of a new sentence and within sentences throughout the manuscript (e.g., beginning of L41, L49, L81, middle of L147, etc.).

·      Check consistency of spacing around hyphens/dashes throughout the manuscript (e.g., L185, no space 4-61; L188, space 0 – 10.33).

·      Check spacing and font color of references.

Line-specific:

·      L31: Add a period at the end of the sentence

·      L49: extra space between utilizes and Google

·      L233: sub in colon after “three-point scale:”

·      L440: there’s an extra space in the first parentheses

·      L452: capitalize “deep” to match the case of the other variables

·      L455: italicize “n” if that is appropriate for the journal formatting (statistical abbreviations are typically denoted in italics)

·      L462: change semi-colon following “alpha” to a comma

·      Figure 9: the percentage values on this chart for some of the bars are a little challenging to read (at a glance). I wonder if changing the orientation (e.g., angling them) might increase readability?

·      L630 onward: Add degree symbols to the temperature data 

·      L661: remove extra period at the end of the sentence

·      L672: remove extra space at the end of the sentence

·      L756: Is this the same Jackson from L447? If so, I would just adjust the naming, because here it is “Jackson-Jewett”

·      L783: the third time reads 13:30:30 – is this correct?

·      L879: check the tab on this opening sentence

·      L880: caution here with the use of “matrix.” An SNA matrix refers to the data itself organized in rows and columns before it is then used to produce a diagram like Figure 14, which is referred to as a sociogram. I would adjust this language in the paragraph as well as the figure description.

·      L884: remove extra period at the end of the sentence

·      L905: Candii is spelled differently in the sentence and in Figure 16

·      L1005: adjust the semi-colon to a comma before “i.e.”

All recommended adjustments pertaining to grammar and mechanics are listed above.

Reviewer 2 Report

This manuscript discusses seven different types of behavioral metrics used to assess chimpanzee welfare. Case studies illustrate how data are used to initiate conversations, and how interventions mitigate concerns. These descriptions and accompanying stories may help institutions that are just developing their programs, or refining them, in identifying which metrics may be helpful and how to collect those data. One additional piece that may be helpful is to include blank templates (or even screenshots) of datasheets in supplementary material, so people can see how the data are organized. I recommend this for two reasons: one is that not every institution is going to have a dedicated behaviorist so any support on the back end would be helpful, and two, since Power Bi is a relatively new software, it would be helpful to see if there are major differences with how one might set up an Excel sheet.

A few line by line comments:

Paragraph beginning at line 52: surely other zoo organizations beyond AZA have similar standards, eg EAZA, WAZA, SEAZA, etc?

Paragraph beginning at line 61: this paragraph unfortunately ignores the body of behavioral research coming from other managed settings, eg laboratory and zoo. I understand the need to keep the scope narrow especially considering the current length of the paper, but identifying great ape research as “uncommon in the past”, while true for sanctuaries due to having not been around as long as other settings, is misleading, and framing it this way may lead the reader to believe these welfare-focused metrics and/or collection methods were innovated in sanctuaries, which they were not.

Line 91: suggest references from Wemelsfelder, Whitham & Wielebnowski, Gosling, etc

Line 99: “objective, evidence-based data” is repetitive

Line 107: Altmann

Lines 144-6: with wellness surveys as frequent as 4 times a year per individual, is seasonal rotation of assessment date necessary? It seems like you would already be capturing all the seasons.

169: suggest ‘enhance’ instead of “increase welfare”

This section on Wellness Surveys begins by suggesting that welfare scores may fluctuate seasonally but there are no data presented to determine this relationship. In the date range included, there are 6, possibly 7, assessments done, so such an analysis, even if preliminary, may be possible.

Lines 197-199: two questions: do you have any data to support intra-observer reliability, and/or inter-observer reliability? It would be interesting to see if an individual assessor’s score changes based on actual measurable changes in the animal’s welfare, or based on changes in the assessor’s understanding of the questions on the surveys. There have been data from San Diego Zoo, Lincoln Park Zoo (and maybe also somewhere in Australia) indicating that scores vary as keepers learn more about welfare, etc, leading some institutions to refine the questionnaire. The follow up question is whether you have other data to support/validate the survey, eg physiological (hormones)? You certainly have plenty of behavioral data!  (See: O’Brien, S. L., & Cronin, K. A. (2023). Doing better for understudied species: Evaluation and improvement of a species-general animal welfare assessment tool for zoos. Applied Animal Behaviour Science264, 105965.   I think this is the Australian one I was thinking of: Sherwen, S. L., Hemsworth, L. M., Beausoleil, N. J., Embury, A., & Mellor, D. J. (2018). An animal welfare risk assessment process for zoos. Animals8(8), 130.)

Lines 372-392: I appreciate the self-reflection and suggestions for improvement. It may help others developing their own datasheets consider which information will be helpful to include.

Line 1003: suggest ‘enhance’ or ‘improve’ instead of “increase”
